# Physical Traits and Reproductive Measurements Associated with Early Conception in Beef Replacement Heifers

**DOI:** 10.3390/ani12151910

**Published:** 2022-07-27

**Authors:** Megan S. Hindman, Brian Huedepohl, Grant A. Dewell, Troy A. Brick, Gustavo S. Silva, Terry J. Engelken

**Affiliations:** 1Veterinary Department of Production Animal Medicine, College of Veterinary Medicine, Iowa State University, Ames, IA 50010, USA; gdewell@iastate.edu (G.A.D.); gustavos@iastate.edu (G.S.S.); engelken@iastate.edu (T.J.E.); 2Veterinary Medical Center, Williamsburg, IA 52361, USA; vmc1@iowatelecom.net; 3Iowa Department of Agriculture and Land Stewardship, Des Moines, IA 50319, USA; troy.brick@iowaagriculture.gov

**Keywords:** beef heifer, replacement heifer, reproductive tract score

## Abstract

**Simple Summary:**

Developing and raising replacement beef heifers requires a large capital investment for producers. Therefore, it is imperative to discover traits and management practices to eliminate subfertile heifers prior to breeding and pregnancy determination. In this study, four years of data was analyzed from a centralized heifer development yard in the Midwest of the United States. The objective of this study was to analyze various heifer physical characteristics and management practices in order to quantify their impact on pregnancy and date of conception. Physical measurements can be used to improve the ability to select and develop heifers for improved reproductive longevity. Veterinarians have an opportunity to work with their clients to utilize this information to select replacements that fit the ranch environment. This should result in increased reproductive efficiency and cost optimization for the replacement heifer enterprise.

**Abstract:**

Developing and raising replacement heifers requires a large capital investment for producers. Therefore, it is imperative to discover traits and management practices to eliminate subfertile heifers prior to breeding and pregnancy determination. In this study, four years of data was analyzed from a centralized beef heifer development yard in the Midwest of the United States. The objective of this study was to analyze various heifer physical characteristics and management practices in order to quantify their impact on pregnancy and date of conception. Logistic regression models were built to investigate risk factors associated with conception to artificial insemination (AI), pregnancy by natural service after AI exposure, and pregnancy in the first 21-days of the breeding season. Age at entry, average daily gain from entry to breeding, pelvic width, and year were associated with AI pregnancy (*p* < 0.05). On the second model, average daily gain from entry to yearling weight, weight at breeding, weight at pregnancy diagnosis, and age at AI were significantly associated with pregnancy. There were no associations with reproductive tract score with any of the response variables analyzed. These results indicate there are physical measurements that can be used to improve the ability to select and develop heifers for improved reproductive performance.

## 1. Introduction

Replacement heifers are a vital component to the maintenance and growth of today’s cattle herds. With an annual replacement rate of 14.9% in the United States, optimal beef heifer development is a critical component to herd profitability [1]. A recent report estimated an average cost of USD 1418 to develop a replacement heifer from weaning to pregnancy determination [2]. The greatest portion of this expense was associated with the opportunity cost of retaining the heifer at weaning [2]. In addition to this expense, the remaining developmental costs were primarily associated with supplemental feed, grazing expenses, and the animal health program. The cost associated with this stage of production is still nearly a year away from weaning a calf and producing income. Combining the average annual replacement rate with the large capital investment, it becomes imperative to select and develop replacement heifers to maintain productive longevity and the long-term success of the cowherd [2]. 

The success in a replacement heifer program can be defined as conceiving in the first 21 days of the breeding season, calving by two years of age with minimum dystocia, and breeding back in a timely fashion [3]. Replacement heifers need to wean three to five calves in order to produce enough revenue stream to offset their initial developmental costs [4]. To promote heifer longevity, heifers need to conceive within 15 months of age and within the first 21 days of the breeding season [5]. Therefore, it is imperative for management decisions to include proper genetic selection, appropriate nutrition, pre-breeding evaluation and selection, and post-breeding management to ensure a live calf and a timely return to estrus for future breeding seasons. 

Fertility traits that influence predictability of puberty in replacement heifers are lowly heritable [6,7]. Therefore, selecting heifers based on parameters important for sustained reproductive performance is critical. Traits producers often select throughout the development process are weight, body condition score (BCS), pelvic measurements, reproductive tract score (RTS), pregnancy status, and genetic potential [8]. Once replacement heifers have been weaned and selected, management strategies are implemented to ensure their proper growth and development prior to breeding. Nutritional management should ensure that replacements reach a target weight of 55 to 65% of expected mature weight prior to the start of their first breeding season [9,10]. Previous studies have shown RTS, BCS, and frame score are useful in selecting for quality replacement heifers in large beef herds in the United States [9,10,11,12]. 

However, there is still a need to better understand what qualities and attributes are required to produce quality replacement heifers in small herds with increased variation to ensure long-term productivity in Midwestern herds. The objective of this study was to assess the relationship between heifer characteristics and management practices with pregnancy status and date of conception in beef herds. 

## 2. Methods and Materials

### 2.1. Study Population

This retrospective observational study utilized production records from 2410 beef heifers housed at a private heifer development yard in Iowa, USA. This facility develops heifers in a confinement environment with both covered and open pens with concrete flooring. Pens remained stagnant throughout the course of the development period. There were 66 different cattle sources from Southeastern Iowa, represented in this data set with heifers born over a four-year period (2014 to 2018). Prior to delivery to the yard, each producer selected the heifers for development based on their operation’s needs and selection criteria. 

### 2.2. Management at Arrival

Heifers arrived at the development yard from November to December each year. Individual identification and needed vaccinations were completed upon entry (Figure 1). 

Heifers were vaccinated with a commercially available modified live virus with bacterin vaccine for Bovine Herpesvirus-1, Bovine Viral Diarrhea virus type 1 and 2, Parainfluenza 3 virus, Bovine Respiratory Syncytial virus, *Campylobacter fetus*, *Leptospira canicola*, *Leptospira grippotyphosa*, *Leptospira hardjo*, *Leptospira icterohaemorrhagiae*, and *Leptospira Pomona* (Bovi-shield Gold^®^ FP^®^ VL5, Zoetis Inc., Kalamazoo, MI, USA) and an intranasal vaccine for Bovine Herpesvirus-1, Parainfluenza 3 virus, and Bovine Respiratory Syncytial virus (Inforce 3^®^, Zoetis Inc., Kalamazoo, MI, USA). Heifers were revaccinated at least 60 days prior to breeding. Owners had the option for heifers to be *Brucella abortus* vaccinated (RB51, Professional Biologics Company, Denver, CO, USA).

Initial body weight, hip height, and temperament score were recorded. Frame score was then determined using the Beef Improvement Federation frame score calculation [13]. Temperament score (1 = calm to 5 = fast, aggressive, and vocalization) was assigned chute side based on exit speed and behavior while in the chute [14]. Heifers were sorted by weight and placed in pens until their end time point.

### 2.3. Nutrition

Throughout the development program, heifers were fed a balanced ration in accordance to NRC guidelines for a developing beef heifer consisting of whole kernel corn, ground hay, corn gluten, and balancer mineral that included an ionophore in fence-line feed bunks [15]. Water was provided ad libitum via automatic waterers in each pen. Heifers were managed to achieve a target of 70% of their expected mature body weight prior to the beginning of the breeding season. Target mature body weight was estimated using the following equation (Target mature body weight = (Frame Score × 75) + 800) × 0.70) [13]. Based on the average arrival weight of the pen and calculated target weight needed to reach puberty, total gain and average daily gain was determined for each pen. Depending upon the average initial weight of the pen, rations were constructed to achieve an average daily gain of 0.68 to 1.0 kg over the course of the developmental program. Body condition score was not collected at any time during the developmental process. 

### 2.4. Pre-Breeding Evaluations

Pre-breeding evaluations were performed by a single veterinarian approximately 30 to 45 days prior to breeding. At this time heifers were weighed, temperament scored, pelvic measured, and had their reproductive tract evaluated. Pelvic height and width were measured rectally using a commercially available caliper (Rice pelvimeter, Lane Manufacturing, Denver, CO, USA). Measurements for both pelvic width and height were read to the nearest half centimeter. Pelvic area (cm^2^) was calculated by multiplying pelvic height and pelvic width. Reproductive tract score (RTS) was assessed using a 1 to 5 scale that evaluated uterine horn diameter, ovarian dimensions, and structures present on the ovaries by manual palpation [16]. An RTS score of =1 indicated a pre-pubertal heifer with no palpable follicles, while a score of 5 indicated the presence of a palpable corpus luteum with large follicles. Heifers were removed from the development yard due to illness, death, an RTS of 1, or due to free-martinism.

### 2.5. Estrus Synchronization and Breeding

Heifers that remained in the program following pre-breeding evaluations underwent estrous synchronization for fixed-time artificial insemination utilizing the 14-day CIDR-PG protocol. A Controlled Internal Drug Release (Eazi-Breed™ CIDR^®^, Zoetis Inc., Kalamazoo, MI, USA) device was inserted vaginally, on day 0. Fourteen days following insertion, the CIDR was removed and on day 30 (16 days after CIDR removal) a two mL intramuscular injection of prostaglandin F2-alpha analog (Estrumate^®^, Merck Animal Health, Summit, NJ, USA) was administered. 

### 2.6. Artificial Insemination

Fixed-time artificial insemination (AI) was completed 66 ± 2 h after prostaglandin injection in all heifers in the group unless they exhibited estrus behavior prior to this time. Heifers that exhibited behavioral estrus early were inseminated 8–12 h after standing estrus. During fixed-time AI, 2 mL of GnRH (Fertagyl^®^, Merck Animal Health, Summit, NJ, USA) was administered intramuscularly. Heifers were inseminated using semen selected by producers and AI’d by a single technician each year. Two days after insemination, Angus clean-up bulls were placed with heifers to complete a 60-day breeding season including fixed-timed AI. A 1:25 bull to heifer ratio was utilized in the pens. These bulls were selected from a single operation and passed a breeding soundness examination based on the Society for Theriogenology guidelines [17]. 

### 2.7. Pregnancy Diagnosis

One experienced veterinarian evaluated pregnancy status in all heifers using trans-rectal ultrasonography (EASI-SCAN, BCF Technology Ltd., Rochester, MN, USA) 80–90 days post insemination. Heifers were classified by days pregnant to determine if pregnant by AI, natural service, or not pregnant. To obtain classification of pregnant by AI, heifers had to be confirmed pregnant within four days from the AI date. Heifers categorized as pregnant by natural service were confirmed pregnant five days or greater from the AI date. 

### 2.8. Statistical Analysis

Among potential explanatory variables described above, only those with sufficient data after the validation were included in the final dataset. All continuous variables in the final dataset were categorized into quartiles, while categorical variables were preserved in the original format. The continuous variables were categorized to improve model interpretation. Two mixed logistic regression models were built to assess potential risk factors associated with the pregnancy status (Yes or No) and date of conception in beef herds. The farm of origin for the heifers was included as random intercept. The final model for each outcome was built using a two-step approach.

First, a univariate analysis was performed separately between each explanatory variable and the outcome. The objective of this step was to identify the set of variables associated with each outcome, where only variables with *p*-values < 0.20 were eligible for inclusion in the initial multivariable model (step two). Before including the variables retained in the first step in the multivariable model, Spearman’s rank correlation coefficients were estimated among all the categoric explanatory significant variables to avoid multicollinearity in the multivariable model. If the correlation between two variables was ≥0.70, one of the two variables was excluded.

To build the multivariable model a manual stepwise backward selection was performed for each model. In summary, all variables with *p* < 0.20 in the univariate analysis were included in the multivariable model and the variables that had *p* > 0.05 were excluded one at a time until obtaining the final model with all variables having *p* < 0.05. Moreover, the Akaike’s Information Criterion (AIC) value was utilized to compare the model goodness of fit when excluding the variables, by targeting the lowest value [18]. Furthermore, confounders and interactions were tested based on the biological plausibility of the variables and literature. Multicollinearity was tested in the final set of predictors included in the multivariable model by measuring the variance inflation factor (VIF) for each parameter and excluding variables with VIF > 5. Furthermore, all predictors not included in the initial multivariable model due to having a *p*-value higher than the cut-off (*p* > 0.20) on the univariate analyses were included again, one at a time, back into the final multivariable model, with the objective of testing whether these variables remain not significant in the presence of potential confounders [19].

To assess the impact of each explanatory variable on the outcome, the odds ratio was computed as the measure of association and the first category for each independent variable (risk factor) was selected as the reference to compute the odds ratio. All analyses were performed using the R program (R Core Team, 2016) and the following packages: *car*, *gtools*, *lme4*, *pROC*, *sjPlot*, *stats*, and *table1* [20].

To assess the impact of each explanatory variable on the outcome, the odds ratio was computed as the measure of association and the first category for each independent variable (risk factor) was selected as the reference to compute the odds ratio. All analyses were performed using the R program (R Core Team, 2016) and the following packages: *car*, *gtools*, *lme4*, *pROC*, *sjPlot*, *stats*, and *table1* [20].

## 3. Results

The initial dataset contained heifer development data for 3418 replacement heifers from 66 different farms born from 2014 to 2018. Heifers were removed from the analysis due to missing data (n = 903), being culled (n = 100), or dying (n = 5) during the developmental period. After the removal, 2410 heifers were included in the analysis. These heifers had an average entry weight and age of 298.63 ± 49.545 kg and 263.40 ± 48.64 days, respectively. These heifers also averaged 371.818 ± 45.909 kg with an overall RTS of 3.22 ± 0.93 at pre-breeding processing. 

Within the dataset, 44% of heifers were confirmed pregnant to AI (n = 1071), with 10.5% of heifers confirmed open after AI and natural service (n = 253). Heifers conceiving to AI averaged 299.091 ± 50.000 kg, and 265 ± 37.4 days of age at entry. These heifers also averaged 373.182 ± 45.909 kg at the time of breeding (Table 1). Average RTS score for heifers confirmed pregnant to AI was 3.218 ± 0.927 (Table 1).

Analysis for AI pregnancy revealed age of entry into the yard, average daily gain from entry to breeding, pelvic width, and age at AI affected this variable (Table 2). The variable year was left in the model to adjust for difference across the years. Logistic regression analysis of the response variable pregnant to natural service revealed average daily gain from entry to breeding, breeding weight, pregnancy check weight, and age at AI affected this variable (Table 3). These results indicate that physical measurements in the heifers in this dataset were associated with pregnancy to AI and pregnancy to natural service. 

The odds of conceiving to AI increased as age of heifers increased (above 266 days) at entry into the yard (*p* < 0.001) (Table 2). This dataset revealed lower odds of conceiving to AI when average daily gain from entry to breeding increased over the initial quartile of 0.84 kg/day to 1.19 kg/day (Table 2) compared to the reference. The odds of conceiving to AI increased when the heifers had pelvic width measures above 15.5 cm compared to heifers with 15 cm or lower (Table 2). 

For the analysis of the response variable pregnancy and explanatory variable ADG from entry to breeding, the heifers that had ADG entry to breeding between 0.39 and 0.62 kg/day had higher odds (1.75, *p* = 0.008) of becoming pregnant compared to the reference (lower than 0.21 kg/day). The odds of a heifer becoming pregnant decreased as breeding weights increased above 372.73 kg compared to heifers with lower than 338.64 kg (Table 3). As weight at pregnancy diagnosis increased, the odds ratio for a heifer to become pregnant increased compared to the reference (below 404.55 kg) (Table 3). Heifers had 0.63 lower odds of becoming pregnant when their age was above 432 days compared to the heifers below 408 days (*p* = 0.032). 

## 4. Discussion

Developing and raising replacement heifers requires a large capital investment for producers. Therefore, it is imperative to discover traits and management practices to eliminate subfertile heifers prior to breeding and pregnancy determination. This will enhance the ability to select heifers for improved reproductive longevity. In this study, data were analyzed from a centralized heifer development yard in the Midwest over a four-year period. Since the owners preselected these heifers prior to entering the heifer development yard, they do not represent a random sample of heifers. However, the findings in this study still provide useful insights into the selection and development of quality beef heifers based on reproductive outcomes.

There was no association between weight at entry into the yard or at pre-breeding with the pregnancy outcomes. The average breeding weight was 371.818 kg, with the average daily gain from entry to breeding of 0.7 kg/day. This corresponds with standards set for replacement heifers of British breeding [21]. It should also be noted that the target body weight of the heifers in this yard was intended to reach 70% of expected mature body weight prior to breeding. Based on the records, this target weight was achieved in most heifers. These results indicate nutritional management plays an integral role in the attainment of puberty and the success of heifer development [8,22,23,24,25]. With a total pregnancy rate of 91.1%, it is conceivable that the relationship of average daily gain and breeding weight influenced pregnancy outcomes in the models in our study [22,26]. This data set also demonstrates that a 70% mature body weight target provided acceptable reproductive performance in these heifers. However, the individual economic impact of feed costs must be considered in order to manage development costs optimally.

An older heifer at entry was more likely to conceive to AI. Research has shown that heifers that are older at weaning tend to attain puberty earlier in their development [5,27]. This will tend to improve conception to AI and other pregnancy outcomes over the entire breeding season. It is conceivable that changes in BCS between arrival and the time of AI could impact this relationship as well. However, because BCS was not recorded at this development yard, we were unable to evaluate the potential effect of body condition changes on reproductive performance.

A heifer with a pelvic width of 15.5 cm to 18 cm had a higher odds of conceiving to AI compared to heifers that had a pelvic width of 15 to 15.5 cm. Puberty exerts a positive influence on pelvic width and pelvic area in yearling heifers [28]. It is important to note that pelvic width increases pelvic area, which in turn increases with the total skeletal structure of the replacement heifer [8]. Pelvic area also positively associated with hip height and overall frame size. By increasing frame score, it also increases the need for higher target breeding weights as well as overall heifer maintenance energy needs and cost. Therefore, although it was significant, it is important to target an individual farm average frame score versus an industry wide standard to achieve the heifer development goals. 

There was no association between RTS and any of the pregnancy outcomes analyzed compared to previous research published [29,30]. A previous study noted RTS was significantly associated with pregnancy outcomes by itself, but was found insignificant when adding additional variables to the model [10]. In this group of heifers, the median days to pregnancy varied by only one day between all RTS. This result could be due to owners selecting heifers prior to entering the yard and innately standardizing their replacement heifers for this trait. In addition, RTS has an estimated heritability of 0.32 [16]. Therefore, herds that have participated in this particular development program over multiple years, have programmed their heifers to attain puberty at an earlier age [16]. This phenomenon would tend to limit the amount of variation in the RTS over multiple generations of heifers and decrease the significance of this variable in the model. 

Based on these findings, there are physical measurements that can be used to improve heifer selection and development practices. This study also shows that producer decisions prior to entering this development yard may have led to selection bias in this population. This bias is unavoidable as producers attempt to select heifers that will fit their operation in order to optimize development costs and reproductive performance at the development yard. Information bias may have also led to discrepancies in the external validity in the results due to absent data and possible lack of accurate measurements. Therefore, these results may not be directly applicable to a producer that retains their entire heifer calf crop as replacements or develops them under a different system. This leaves an opportunity for the producer and their veterinarian to collect and analyze data to better select heifers that will conceive early in the breeding season and excel in productive longevity. 

## 5. Conclusions

There are physical measurements that can be used to improve the ability to select and develop heifers for improved reproductive longevity. Increased age at entry into the developmental yard, increased pelvic width, ADG throughout the development period to include 0.5 kg/day, and lower age at AI are all factors that influenced the odds of conception to AI. RTS was not predictive in the pregnancy outcomes analyzed. Veterinarians have an opportunity to work with their clients to utilize this information to select replacements that fit the ranch environment. This should result in increased reproductive efficiency and cost optimization for the replacement heifer enterprise. 

## Figures and Tables

**Figure 1 animals-12-01910-f001:**
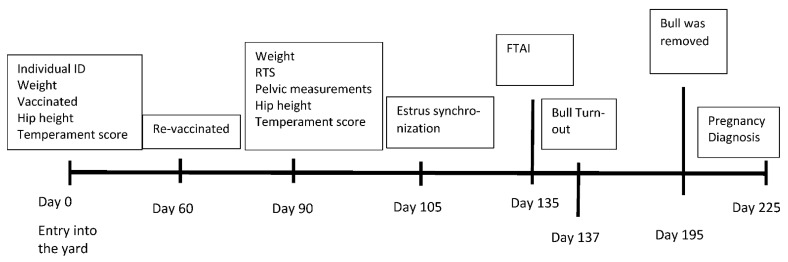
Timeline of measurements and events for the heifer development yard. RTS—reproductive tract score, FTAI—Fixed timed AI.

**Table 1 animals-12-01910-t001:** Descriptive statistics of replacement beef heifers (AI pregnant vs. all other heifers).

Variable	AI Pregnant Heifers	All Other Heifers
	Mean (S.D)	Mean (S.D.)
Entry Weight (kgs)	299.09 (50.00)	298.18 (48.64)
Age at Entry	265 (37.4)	262 (35.6)
Breeding Weight (kgs)	373.18 (45.91)	370.91 (45.91)
Reproductive Tract Score	3.22 (0.93)	3.22 (0.93)

**Table 2 animals-12-01910-t002:** Results of the final multivariable logistic regression model for AI.

Risk Factors	Categories ^†^	Odds Ratio	CI 95% ^‡^	*p*-Value
*Year*	2014	*Ref **	*Ref*	*Ref*
	2015	5.97	4.13–8.61	<0.01
	2016	9.05	6.00–13.64	<0.01
	2017	6.85	4.67–10.05	<0.01
	2018	7.22	4.84–10.77	<0.01
*Age at entry (days)*	133–243	*Ref*	*Ref*	*Ref*
	243.1–266	0.98	0.74–1.31	0.90
	266.1–289	2.04	1.52–2.74	<0.01
	289.1–415	1.83	1.33–2.47	<0.01
*ADG Entry to Breeding (kg/day)*	−5.0–0.84	*Ref*	*Ref*	*Ref*
	0.85–1.19	1.32	1.02–1.69	0.03
	1.20–1.66	1.00	0.76–1.31	0.84
	1.67–10.0	0.92	0.66–1.27	0.58
*Pelvic Width (cm)*	4–15	*Ref*	*Ref*	*Ref*
	15.1–15.5	0.91	0.73–1.14	0.42
	15.6–16	1.48	1.18–1.86	0.01
	16.1–18	1.60	1.14–2.23	<0.01
*Age at AI (days)*	302–408	*Ref*	*Ref*	*Ref*
	408.1–421	0.74	0.56–0.98	0.04
	421.1–432	0.55	0.41–0.74	<0.01
	432.1–599	0.49	0.37–0.65	<0.01

^†^ Categories = quantiles of the categoric variables. ^‡^ CI 95% = confidence interval at 95%. * Ref = reference category for each independent variable. Bold *p* values are signficant explanatory variables for the logistic regression model.

**Table 3 animals-12-01910-t003:** Results of the final multivariable logistic regression model for pregnancy.

Risk Factors	Categories ^†^	Odds Ratio	CI 95% ^‡^	*p*-Value
*ADG Entry to Yearling Weight (kg/day)*	−1.3–0.46	*Ref **	*Ref*	*Ref*
	0.47–0.85	0.88	0.62–1.27	0.50
	0.86–1.36	1.69	1.12–2.56	<0.01
	1.37–6.67	1.27	0.85–1.89	0.25
*Breeding Weight (kg)*	235.00–338.64	*Ref*	*Ref*	*Ref*
	338.65–372.73	0.7	0.45–1.08	0.11
	372.74–404.55	0.56	0.33–0.93	0.02
	404.56–545.46	0.41	0.23–0.73	<0.01
*Pregnancy Diagnosis Weight (kg)*	227.27–404.55	*Ref*	*Ref*	*Ref*
	404.56–438.64	1.25	0.83–1.89	0.29
	438.65–472.73	1.87	1.16–3.01	0.01
	472.74–609.09	2.49	1.41–4.40	<0.01

^†^ Categories = quantiles of the categoric variables. ^‡^ CI 95% = confidence interval at 95%. * Ref = reference category for each independent variable. Bold *p* values are signficant explanatory variables for the logistic regression model.

## Data Availability

The data presented in this study are available on request from the corresponding author. The data are not publicly available due to privacy of the heifer development yard and its producers.

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
