# Peer review of "Physical Traits and Reproductive Measurements Associated with Early Conception in Beef Replacement Heifers"

_animals, 2022, doi:10.3390/ani12151910_

Round 1

Reviewer 1 Report

The present manuscript reports associating physical and reproductive measurements with conception in beef heifers. The data were from 2416 heifers in a heifer-raising facility over four years. The objective was to determine if physical attributes could be used to select heifers for the likelihood of becoming pregnant (L247). Animals ranged from 133 to 415 days of age at delivery to the facility. Age, weight, hip height, and temperament were recorded at entry. Animals were enrolled in estrous synchronization with a CIDR 105 days later, and similar data were collected in addition to reproductive tract score and pelvic measurements. The CIDR was removed after 14 days, and prostaglandin injections were given16 days later (Day 135). Heifers were inseminated 66 hours later (fixed-time artificial insemination), followed by the bull introduction, which was removed at 195 days. The pregnancy diagnosis was on Day 225. The investigators examined for effects of age at the various events on pregnancy; however, the results of ages are not independent. For example, an animal that began at 200 days of age is 290 at pre-breeding, 305 at estrous synchrony, and 337 at breeding, and one that is 400 would be 490 at pre-breeding, 505 at estrous synchrony, and 537 at breeding. Thus, animal distributions into each quartile will be close to the same - shared variance. Therefore, only one age, probably breeding age, would be necessary for prediction, as is reflected in the models, because all animals are in a fixed program after entry. But, what do we learn from that – many animals will be prepubertal at 10 months of age at FTAI. In Table 3, the fourth quartile for Age at AI is significant because older heifers (up to 20 months) likely were infertile to prior services before entering the facility. [Analysis of a subset of data at 13-15 months might answer the producer’s question- What variables predict fertility in animals when breeding at ages to yield calving at 24 months? ] 

Variables examined at pregnancy diagnosis are after the fact; pregnancy status is known then. They will not help predict fertility. Interestingly, pelvic width was important as a single variable but did not fit the multivariable model, and the reproductive tract score was not usable for predicting pregnancy. The pelvic score would lend itself to pregnancy delivery outcomes. Temperament was not a predictive factor.

Questions about the data:

The age range at entry was 133 to 415; the range at breeding 135 days later was 302 to 599. Should the latter range be 268 to 550? If the decision was not to breed animals under 10 months, then data for those animals should be removed from all data sets.

ADG from entry to breeding for one animal is -5 kg/day, loss of 5*135 days= 675kg.

ADG from entry to yearling for one is -82.5 kg/day.

On pen-feeding to result in ADG ~0.68 to 1 kg, animal(s) lost weight between breeding and pregnancy diagnosis (90 days).

L 14 here and throughout data (plural) are

L 16 and 33 delete “There are” and “that”

L 38 delete “today’s”

L 82 delete “individual”

L 99 add a marker at Day 195 on the timeline for the bull was removed

L 154 GnRH

L 157 Is this a 60-day season including FTAI or AI to estrus?

L 173 delete space after herds

L 181 change parameters to variables

L 206 Here and Table 1 use significant digits

L 206, 207, 210 include SE

L 213 include the number of animals in Table 1

Table 2, The last number in the quartile should be distinct from the first in the ensuing quartile. For example, 243 cannot end the first quartile and begin the second quartile for age at entry.

L 233 compared

L 248 delete “and pregnancy determination”  The goal is to predict fertile heifers. What is monitored at pregnancy diagnosis is not a predictor of fertility.

L 249 – 254  Move to summary or eliminate

L 269 delete “at this time”

L 271 delete “phase”

L 287 remove the indent

Author Response

  • The present manuscript reports associating physical and reproductive measurements with conception in beef heifers. The data were from 2416 heifers in a heifer-raising facility over four years. The objective was to determine if physical attributes could be used to select heifers for the likelihood of becoming pregnant (L247). Animals ranged from 133 to 415 days of age at delivery to the facility. Age, weight, hip height, and temperament were recorded at entry. Animals were enrolled in estrous synchronization with a CIDR 105 days later, and similar data were collected in addition to reproductive tract score and pelvic measurements. The CIDR was removed after 14 days, and prostaglandin injections were given16 days later (Day 135). Heifers were inseminated 66 hours later (fixed-time artificial insemination), followed by the bull introduction, which was removed at 195 days. The pregnancy diagnosis was on Day 225. The investigators examined for effects of age at the various events on pregnancy; however, the results of ages are not independent. For example, an animal that began at 200 days of age is 290 at pre-breeding, 305 at estrous synchrony, and 337 at breeding, and one that is 400 would be 490 at pre-breeding, 505 at estrous synchrony, and 537 at breeding. Thus, animal distributions into each quartile will be close to the same - shared variance. Therefore, only one age, probably breeding age, would be necessary for prediction, as is reflected in the models, because all animals are in a fixed program after entry. But, what do we learn from that – many animals will be prepubertal at 10 months of age at FTAI. In Table 3, the fourth quartile for Age at AI is significant because older heifers (up to 20 months) likely were infertile to prior services before entering the facility. [Analysis of a subset of data at 13-15 months might answer the producer’s question- What variables predict fertility in animals when breeding at ages to yield calving at 24 months? ] 
    • Thank you for your comment, while we agree with your logic, the analysis of a subset of data was not the objective of this study. We re-ran the model and age at AI was not a removed from it due to lack of signficance. Model was adjusted in the manuscript.
  • Variables examined at pregnancy diagnosis are after the fact; pregnancy status is known then. They will not help predict fertility. Interestingly, pelvic width was important as a single variable but did not fit the multivariable model, and the reproductive tract score was not usable for predicting pregnancy. The pelvic score would lend itself to pregnancy delivery outcomes. Temperament was not a predictive factor.
    • The objective of the study was to conduct a retrospective observational study to identify risk factors. Retrospective studies are useful to generate hypothesis by identify factors that can be investigated later. The objective of the study was not to build a prediction model. In addition, when building prediction models, you always need the outcome to be able to train the model, so you need the result of the outcome either way. The authors respectfully disagree with the reviewer and didn’t understand the point the reviewer is making.
  • The age range at entry was 133 to 415; the range at breeding 135 days later was 302 to 599. Should the latter range be 268 to 550? If the decision was not to breed animals under 10 months, then data for those animals should be removed from all data sets.
    • All heifers regardless of age were included into this model. It is likely they did not appear in the dataset due to owner selecting for older heifers prior to entering the development yard. Thus we have included a bias statement in the discussion stating there was selection bias from the owners due to economical reasons of entering the yard and being successful. The only reasons animals removed or not included in the dataset were due to mortality, culled (free-martinism or steers (woops)), or missing data (no pelvic measurements, weights, etc.).
  • ADG from entry to breeding for one animal is -5 kg/day, loss of 5*135 days= 675kg.
    • There was a misunderstanding. This variable reflects the ADG for the entire period.
  • ADG from entry to yearling for one is -82.5 kg/day.
    • There was a typo in the table and was corrected.
  • On pen-feeding to result in ADG ~0.68 to 1 kg, animal(s) lost weight between breeding and pregnancy diagnosis (90 days).
    • We did not set up an exclusion criteria for this dataset as it is analyzing a "real-time" yard retrospectively, and by excluding weight loss as a criteria we did not depict a representation of what could or actually happened in development yards. 
  • L 14 here and throughout data (plural) are
    • This has been fixed.
  • L 16 and 33 delete “There are” and “that”
    • This has been fixed.
  • L 38 delete “today’s”
    • This has been fixed.
  • L 82 delete “individual”
    • This has been fixed.
  • L 99 add a marker at Day 195 on the timeline for the bull was removed
    • This has been fixed.
  • L 154 GnRH
    • This has been fixed.
  • L 157 Is this a 60-day season including FTAI or AI to estrus?
    • This has been fixed.
  • L 173 delete space after herds
    • This is a formatting issue from the publisher, and the author is unable to fix it at this time. Maybe the publisher/editor can?
  • L 181 change parameters to variables
    • This has been fixed.
  • L 206 Here and Table 1 use significant digits
    • This has been fixed.
  • L 206, 207, 210 include SE
    • This has been fixed.
  • L 213 include the number of animals in Table 1
    • Please see above in results section, Line 215-216
  • Table 2, The last number in the quartile should be distinct from the first in the ensuing quartile. For example, 243 cannot end the first quartile and begin the second quartile for age at entry.
    • Tables 2 and 3 were corrected to address this comment.
  • L 233 compared
    • This has been fixed. 
  • L 248 delete “and pregnancy determination”  The goal is to predict fertile heifers. What is monitored at pregnancy diagnosis is not a predictor of fertility.
    • The objective of the study was to conduct a retrospective observational study to identify risk factors. Retrospective studies are useful to generate hypothesis by identify factors that can be investigated later. The objective of the study was not to build a prediction model. In addition, when building prediction models, you always need the outcome to be able to train the model, so you need the result of the outcome either way. The authors respectfully disagree with the reviewer and didn’t understand the point the reviewer is making.
  • L 249 – 254  Move to summary or eliminate
    • The author is unclear what the reviewer meant. 
  • L 269 delete “at this time”
    • This has been changed.
  • L 271 delete “phase”
    • this has been changed
  • L 287 remove the indent
    • This is a formatting issue from the publisher, and the author is unable to fix it at this time. Maybe the publisher/editor can?
    •  

Reviewer 2 Report

This reviewer appreciates the amount of effort it took to collect and analyse the data. It includes some practical information, but it was not designed in an experimental method that would have been more beneficial. This was beccause it is an "after the fact" data set and there were many things that the authors could not change. 

It would be very useful to know more about these heifers. For example, what were their breeds or were they mostly crossbreds? Did they include beef and dairy breeds or crossbreds of beef and dairy breeds? How many heifers arrived on any day?  Were there any adjustments made for climatic conditions (doors, gates, fans, etc)? Were the heifers all fed the same ration throughout all of the five years, regardless of age of heifers. Are there any data on the general nutritional analyses of the feed? 

What is meant by a "balanced" ration? How much energy and protein was provided per animal at various weights? What amount was fed per animal, and did this change with changes in weather conditions? Was the corn fed in the ration ground? If so, what size of grind? What is meant by ground hay? What kind of hay and what length of chop or grind? Was the 70% of mature body weight before breeding based on the apparent breed of the heifers or was this an overall average? What was the average target weight?

The authors state that heifers were in buildings and on concrete. Was there any adjustment for square meters per heifer as heifers grew in size and weight? If they were fed at a feed bunk, was their adequate feed available for all heifers as needed? What type of bedding and housing accomodations did the heifers have? Were they fed more during extreme cold weather? 

It is not clear how the authors determined that a heifer conceived to timed AI or to the bull, because there was only 2 days between the end of timed AI and entry of the bull. If all heifers responded similarily to the timed AI program, then bull-conceived fetuses would be smaller; however, many published studies have shown that timed-AI programs have an "accurate" response rate of around 80%. So how was this accounted for?

It is not clear if the heifers stayed in the same pen from entry until leaving. Obviously the amout of space per heifer would change as they grew larger. What was the average pen size? Was pen size considered in the analyses of data?

A major issue that is not addressed in this manuscript is why the probability of becoming pregnant by artificial insemination was significantly greater during years 2015 through 2018 than during 2014 (the base year). Did the facility begin this heifer-breeding program in 2014? It is important to know what differences occurred during 2015-2018 to improve pregnancy to AI. Otherwise, it is challenging to understand how other factors such as age or weight influenced fertility and pregnancy rate. 

If the authors used season of year rather than year as the entry time, the data may have shown clearer effects of spring-calving vs fall-calving or effects of summer heat or winter cold on traits measured. Clearly there are more differences among seasons than among years. 

Another issue is what is new or informative from this study? It was not a designed experiment, but it has a large number of observations. So what is the "take-home" message that was not already known before these data were analyzed?

L147 change to 2 mL

L148 change to a Prostaglandin F2-alpha analog (Estrumate)

L154 change to GnRH

L164 It does not seem feasible that pregnancy diagnosis at 80-90 days post-insemination could distinguish between a heifer that conceived by AI or by a bull that bred her 2 days later. Explain how this was done. 

L168 described

L175 built

L194 What is meant by a "large P value"?

L198 It is not always clear what is meant by the first category for each variable. Please define more clearly. 

L202 Approximately 30% of the original heifers were excluded from the analyses. This could create a significant bias in the final set of heifers. What was the major reason for removal? If it was missing data, why was it missing? What percentage was associated with serious disease or death? Were heifers that failed to conceive included in the final analyses?

L210 and elsewhere. Do not include decimals when the primary numbers are large. Just state 299+/-50

Tables 2 and 3 are very confusing. The Category columns are confusing, particulary when they have two different values and some have brackets on both ends and some have parentheses and brackets. It is not clear what these mean. 

Table 2. Why are the AI pregnancy rates considerably greater in 2015-2018 than in 2014? It seems that these differences among years reflects something this is missing in the analyses. Different Ai technician? Improved experience in managing AI? Was the same Timed AI program used each year? How do the authors explain this difference?

Table 3 seems to show that a heifer lost 82.5 kg per day (first line [-82.5, 0.46]. What does this represent? 

Author Response

  • This reviewer appreciates the amount of effort it took to collect and analyse the data. It includes some practical information, but it was not designed in an experimental method that would have been more beneficial. This was beccause it is an "after the fact" data set and there were many things that the authors could not change. 
  • It would be very useful to know more about these heifers. For example, what were their breeds or were they mostly crossbreds? Did they include beef and dairy breeds or crossbreds of beef and dairy breeds? How many heifers arrived on any day?  Were there any adjustments made for climatic conditions (doors, gates, fans, etc)? Were the heifers all fed the same ration throughout all of the five years, regardless of age of heifers. Are there any data on the general nutritional analyses of the feed? 
    • The author appreciates the comment. Throughout the manuscript it is stated this is based on beef heifers only. Breeds were not recorded in the dataset however, this would be covered in the farm effect of the statistical analysis as most farms do not own multiple breeds of cattle. All heifers arrived at once and the weather is the same throughout the development period since they entered the yard at once November-December and left all at once at the end of the breeding season as depicted by figure 1. They were fed all the same ration and it was fed in accordance with NRC guidelines as stated in the manuscript. The average target weight was determined on a per head/per farm basis and was not used as a one-size-fits-all per literature review. This would also be embedded within the farm effect in the statistical model and would be a cofounder to the weights actually measured therefore, it was excluded in our actual analysis.
    • The author appreciates the comment
  • What is meant by a "balanced" ration? How much energy and protein was provided per animal at various weights? What amount was fed per animal, and did this change with changes in weather conditions? Was the corn fed in the ration ground? If so, what size of grind? What is meant by ground hay? What kind of hay and what length of chop or grind? Was the 70% of mature body weight before breeding based on the apparent breed of the heifers or was this an overall average? What was the average target weight?
    • The ration was balanced via NRC guidelines. Protein and energy were not calculated daily, weekly, quarterly, or monthly. This falls out of the scope of the study as this is not a nutritional-based paper such as research published by Funston etc, this is a retrospective analysis looking at a heifer development yard and analyzing measurements within the period to predict pregnancy. Ration would be confounding to weights, and could argue as a confounder to fertility as stated in previous work completed by Funston.
    • Again target weight was not analyzed in this model as it would be a confounder with weights actually measured throughout this study.
  • The authors state that heifers were in buildings and on concrete. Was there any adjustment for square meters per heifer as heifers grew in size and weight? If they were fed at a feed bunk, was their adequate feed available for all heifers as needed? What type of bedding and housing accomodations did the heifers have? Were they fed more during extreme cold weather? 
    • Thank you for your comment. In current beef cattle productions, pen adjustments for square footage are based on target end weights and size. Therefore the square footage was already taken into consideration throughout the development period as the pen size met the final square footage requirements for the development period. Pens also remained stagnant meaning no heifers moved from pen to pen throughout the development period. This has been stated in the manuscript Line 81. Bedding and housing accomodations were stated in confinement with concrete floors Line 80. The heifers were not fed more during an extreme cold environment and all heifers were acclimated prior to based on normal seasonal changes. 
  • It is not clear how the authors determined that a heifer conceived to timed AI or to the bull, because there were only 2 days between the end of timed AI and the entry of the bull. If all heifers responded similarly to the timed AI program, then bull-conceived fetuses would be smaller; however, many published studies have shown that timed-AI programs have an "accurate" response rate of around 80%. So how was this accounted for?
    • Thank you for the comment. It is stated that a single veterinarian over all four years determine gestational age using ultrasonography lines 165-166. in Lines 167 it states how retrospectively they were calculated as pregnant to AI or natural service. 80% is a lofty, small research study with uniform herd of multiparous cows. Most bos taurs research articles state a average AI conception rate on Fixed time AI results (being the key) as 40-60% with heifer lower. The gestational age was used to determine both at the time of ultrasounding therefore the fetus would be smaller on the ultrasound image. AI conception rate for this manuscript is 44% stated in line 215.
  • It is not clear if the heifers stayed in the same pen from entry until leaving. Obviously the amout of space per heifer would change as they grew larger. What was the average pen size? Was pen size considered in the analyses of data?
    • Please see comment above about pen space allotment and stagnant pen. 
  • A major issue that is not addressed in this manuscript is why the probability of becoming pregnant by artificial insemination was significantly greater during years 2015 through 2018 than during 2014 (the base year). Did the facility begin this heifer-breeding program in 2014? It is important to know what differences occurred during 2015-2018 to improve pregnancy to AI. Otherwise, it is challenging to understand how other factors such as age or weight influenced fertility and pregnancy rate. 
    • Thank you for your comment, it is a good point you make. The facility did begin heifer breeding program prior to 2014, however most of the data we were wanting to have was missing from their recorded data. There were no management changes throughout 2014-2018. Weather could have been a factor, types of cattle, and owners are all factors that would change. Thus we included year as a confounder understanding that we would not be able to account for all of the year differences. 
  • If the authors used season of year rather than year as the entry time, the data may have shown clearer effects of spring-calving vs fall-calving or effects of summer heat or winter cold on traits measured. Clearly there are more differences among seasons than among years. 
    • There would be no differences between seasons in this data set as they were only developed between November-August each year. No heifers were started or stopped at a different time. Therefore no seasonal effect based on pregnancy predictors and outcomes. 
  • Another issue is what is new or informative from this study? It was not a designed experiment, but it has a large number of observations. So what is the "take-home" message that was not already known before these data were analyzed?
    • Thank you for your comment. Observational studies have the objective to identify factors that can be tested in a controlled study when adjusting by other potential confounders.
  • L147 change to 2 mL
    • It has been changed
  • L148 change to a Prostaglandin F2-alpha analog (Estrumate)
    • It has been changed
  • L154 change to GnRH
    • It has been changed
  • L164 It does not seem feasible that pregnancy diagnosis at 80-90 days post-insemination could distinguish between a heifer that conceived by AI or by a bull that bred her 2 days later. Explain how this was done. 
    • Thank you for the comment. It is stated that a single veterinarian over all four years determine gestational age using ultrasonography lines 165-166. in Lines 167 it states how retrospectively they were calculated as pregnant to AI or natural service.
  • L168 described
    • It has been changed
  • L175 built
    • It has been changed
  • L194 What is meant by a "large P value"?
    • Thanks for catching this mistake. We meant to say p > 0.20. The text was modified to address your comment.
  • L198 It is not always clear what is meant by the first category for each variable. Please define more clearly. 
    • Thanks for your comment. We modified the text to address your comment and the following wording was placed instead “the first category for each independent variable (risk factor) was selected as the reference to calculate the odds ratio”.
  • L202 Approximately 30% of the original heifers were excluded from the analyses. This could create a significant bias in the final set of heifers. What was the major reason for removal? If it was missing data, why was it missing? What percentage was associated with serious disease or death? Were heifers that failed to conceive included in the final analyses?
    • Thank you for your comment, the numbers have been included into the manuscript in line 210 behind the reason.
  • L210 and elsewhere. Do not include decimals when the primary numbers are large. Just state 299+/-50
    • This has been adjusted. 
  • Tables 2 and 3 are very confusing. The Category columns are confusing, particularly when they have two different values and some have brackets on both ends and some have parentheses and brackets. It is not clear what these mean. 
    • Thanks! Tables 2 and 3 were modified to address your suggestion.
  • Table 2. Why are the AI pregnancy rates considerably greater in 2015-2018 than in 2014? It seems that these differences among years reflects something this is missing in the analyses. Different Ai technician? Improved experience in managing AI? Was the same Timed AI program used each year? How do the authors explain this difference?
    • Thank you, First, we decided to include Year in the model to adjust for confounding since we are looking for data across multiple years. Every protocol was the same throughout the years of this dataset as stated in the methods and materials. A single AI technician was also used as stated in the manuscript.
  • Table 3 seems to show that a heifer lost 82.5 kg per day (first line [-82.5, 0.46]. What does this represent? 
    • Thanks for catching this mistake. The typo was corrected.

Reviewer 3 Report

Authors performed a very interesting, retrospective study which tried to explain how different parameters could be effective for heifer selection. The manuscript is well written, but I have some issues prior to publishing the article.

Major issues:

- I would recommend authors to improve the figures and tables, as well, to include more data results in the results section.

- They should point out that as it is a retrospective study, the data obtained could have some gaps or issues.

Some other comments:

- ln 4. avoid DVM, MS, etc.

- ln 6-7. missing author acronyms and e-mails.

- ln 13 change determination by diagnosis (here and throughout the manuscript)

- ln 14 midwest of USA, I guess...

- ln 15 retrospectively?

- abstract: include the parameters analyzed. The conclusion should be improved.

- ln 40: "[1]." Change it throughout the manuscript.

- ln 44 associated with

- ln 78. USA

- Study populaiton. Provide breeds and number of heifers per breed. As well, ages, means, etc.

- Figure 1.  I encourage authors to improve this figure.

- ln 120. Include something like: following the NRC nutrition suggestion, or something like that; and the NRC reference.

- ln 138-139. did not you use ultrasound sncanner?

- ln 156. number of bulls? I guess it was frozen, is not it? Breeds? sex shorted?

- ln 165. four days from the AI date? I doubt about the meaning. Would you like to say that the embryo have to be day compatible with a variation of 4 days? Please rephrase. The next sentence is so much clear than this one. Here you could repeat that the bull was introduced 2d after AI.

- ln 202. in the ln 77 you said another different number, which is also different from ln 203

- ln 203 and 210. days +- sd

- ln 208. 44+10.5% = 50.5%. the rest of heifers?

- in general, 3 decimals are not necessary

- table 2. Could you explain better this table? I do not understand, for example, if the year is a risk factor for AI or natural service (ln 218). The OR would like to mean the times which is better or worst for the conception? Please explain as well table 3. In the next paragrpah is properly explained, but the tables should be understandable by itself.

- some referencs are incomplete, please review

Author Response

Authors performed a very interesting, retrospective study which tried to explain how different parameters could be effective for heifer selection. The manuscript is well written, but I have some issues prior to publishing the article.

Major issues:

  • I would recommend authors to improve the figures and tables, as well, to include more data results in the results section.
    • The author has made corrections to fix figures and tables in the results section
  • They should point out that as it is a retrospective study, the data obtained could have some gaps or issues.
    • Thank you for the comment, the retrospective observational study is stated in line 78 in materials and methods section.

Some other comments:

  • ln 4. avoid DVM, MS, etc.
    • Thank you for the comment: The author will default to the editor if they chose to include or leave out DVM/MS titles
  • ln 6-7. missing author acronyms and e-mails.
    • Thank you for the comment, it is unknown what acronyms are missing and what emails are as the corresponding author has an email listed in line 10.
  • ln 13 change determination by diagnosis (here and throughout the manuscript)
    • Completed
  • ln 14 midwest of USA, I guess...
    • It has been changed
  • ln 15 retrospectively?
    • This is stated in Methods and Materials Line 78.
  • abstract: include the parameters analyzed. The conclusion should be improved.
    • Parameters analyzed can be found in lines 27-30 for the predictive models. Based on the Journals word count, we are unable to provide a complete list of parameters analyzed.
  • ln 40: "[1]." Change it throughout the manuscript.
    • The author is unsure what is meant by "[1]". Can you please re-state what you want changed, is it the period location, the []. The editor requested to change to [] formatting prior to reviewers. 
  • ln 44 associated with
    • Has been changed
  • ln 78. USA
    • Has been changed
  • Study populaiton. Provide breeds and number of heifers per breed. As well, ages, means, etc.
    • There were no breeds recorded in the data set given, the farm effect in the statistical model would account for the difference in breeds, number of heifers per farm, and their average age based on normal farm practices.
  • Figure 1.  I encourage authors to improve this figure.
    • Thank you for the comment, the formatting changed when it went from word document to "manuscript printed" from the editor. The author has made some changes to that, let me know if you need more specific things adjusted. 
  • ln 120. Include something like: following the NRC nutrition suggestion, or something like that; and the NRC reference.
    • The author has made the change.
  • ln 138-139. did not you use ultrasound sncanner?
    • The author has made the change
  • ln 156. number of bulls? I guess it was frozen, is not it? Breeds? sex shorted?
    • This would fall out of the scope of the project as the objective is looking at predictive factors of the development period of beef heifers. The authors did place that it was determined by the owner of the heifers and they were AI'd by a single technician each year.
  • ln 165. four days from the AI date? I doubt about the meaning. Would you like to say that the embryo have to be day compatible with a variation of 4 days? Please rephrase. The next sentence is so much clear than this one. Here you could repeat that the bull was introduced 2d after AI.
    • Thank you for your comment, the author is unsure of what you mean by this comment. Can you please restate what you mean? Utilizing common ultrasound gestation age charts, it is common to be within a standard deviation of 2-4 days from the gestation age.
  • ln 202. in the ln 77 you said another different number, which is also different from ln 203
    • This has been corrected.
  • ln 203 and 210. days +- sd
    • This has been corrected.
  • ln 208. 44+10.5% = 50.5%. the rest of heifers?
    • Correct, 50.5% would be heifers that are not conceived via AI and have been conceived via natural service or Bulls after artificial insemination. 
  • in general, 3 decimals are not necessary
    • Thanks for your suggestion. The manuscript was modified to report the number with 2 decimals.
  • table 2. Could you explain better this table? I do not understand, for example, if the year is a risk factor for AI or natural service (ln 218). The OR would like to mean the times which is better or worst for the conception? Please explain as well table 3. In the next paragrpah is properly explained, but the tables should be understandable by itself.
    • Thanks for your suggestion. We modified the two tables to make them self-explanatory. The variable Year was included only in the model for AI. Year was left in the model to adjust for possible confounding. The interpretation of the estimates (odds ratio) are the following: if the odds ratio is higher than 1 this suggest that the odds of AI conception increase, if it is lower than 1 the odds of AI conception decrease compared to the reference category. The reference category for the analyses was set up to be the first because this increase the model interpretation and assessment of linear increase or decrease in the odds ratio.
  • some referencs are incomplete, please review
    • These have been corrected.

Round 2

Reviewer 1 Report

ADG from entry to breeding ("for the entire period") is -5kg/day. The period is 135 days, which means that the animal lost 675 kg from entry to breeding. Did that seem excessive? Breeding weights range from 235-545 kg.

The point about weight at pregnancy determination (1-3 months after conception) is that it is not a valued predictor of fertility. The additional expenses from FTAI have been incurred for 3 months. Åžentürklü S, Landblom D, Perry G, Petry T. (Effect of Heifer Frame Score on Growth, Fertility, and Economics Anim Biosci 2015;28(1):69-78. DOI: https://doi.org/10.5713/ajas.13.0833):

Breeding weight is the target. The animals in your study were fed to attain 70% mature BW at the beginning of the breeding season, which follows Freetly HC, Kuehn LA, Cundiff LV (2011, Growth curves of crossbred cows sired by Hereford, Angus, Belgian Blue, Brahman, Boran, and Tuli bulls, and the fraction of mature body weight and height at puberty. J Anim Sci 89:2373–2379) that attainment of an absolute BW, or age, was less critical than developing heifers across breeds to a proportion of mature BW. Further, Kasimanickam RK, Kasimanickam VR, McCann ML. Difference in Body Weight at Breeding Affects Reproductive Performance in Replacement Beef Heifers and Carries Consequences to Next Generation Heifers. Animals (Basel). 2021 Sep 26;11(10):2800. doi: 10.3390/ani11102800. PMID: 34679822; PMCID: PMC8533008.

In conclusion, 55% of MBW at breeding negatively affected the reproductive performance of heifers and its offspring heifers. The recommendation is to feed heifers a balanced diet to reach 65% of MBW at breeding with consideration of production traits.

The cattle in your study are of various breeding.  So, would it not be of more importance for the modeling to be based on the percentage of mature BW rather than kg.

L 21 “ to discover traits and management practices to eliminate subfertile heifers prior to breeding” implies indirect selection for the counter: fertile heifers. Your reply-

The objective of the study was to conduct a retrospective observational study to identify risk factors. Retrospective studies are useful to generate hypothesis by identify factors that can be investigated later. The objective of the study was not to build a prediction model. In addition, when building prediction models, you always need the outcome to be able to train the model, so you need the result of the outcome either way.

None of the risk factors that you examined is novel (e.g. Moorey, S.E., Biase, F.H. Beef heifer fertility: importance of management practices and technological advancements. J Animal Sci Biotechnol 11, 97 (2020). https://doi.org/10.1186/s40104-020-00503-9). Further,  a comparable study, Dickinson SE, Elmore MF, Kriese-Anderson L, et al. (Evaluation of age, weaning weight, body condition score, and reproductive tract score in pre-selected beef heifers relative to reproductive potential. Journal of Animal Science and Biotechnology. 2019 ;10:18. DOI: 10.1186/s40104-019-0329-6. PMID: 30891236; PMCID: PMC6390375), which you cited, had reported

Here, we tested the hypothesis that in a group of pre-selected heifers, records of weaning weight, age at weaning, age at artificial insemination, and age of dam differ among heifers of varied reproductive outcomes during the first breeding season. None of the parameters tested presented predictive ability to discriminate the heifers based on the response variable ('pregnant to artificial insemination', 'pregnant to natural service', 'not pregnant'). Heifers younger than 368 d at the start of the breeding season did not become pregnant to artificial insemination. Those young heifers had 12.5% chance to become pregnant in their first breeding season, compared to 87.5% if the heifers were older than 368 days.

Author Response

  1. ADG from entry to breeding ("for the entire period") is -5kg/day. The period is 135 days, which means that the animal lost 675 kg from entry to breeding. Did that seem excessive? Breeding weights range from 235-545 kg.
    1. Thank you for your comment, the authors do agree that it does seem excessive. That particular heifer would be considered an outlier in the data set. 

The point about weight at pregnancy determination (1-3 months after conception) is that it is not a valued predictor of fertility. The additional expenses from FTAI have been incurred for 3 months. Åžentürklü S, Landblom D, Perry G, Petry T. (Effect of Heifer Frame Score on Growth, Fertility, and Economics Anim Biosci 2015;28(1):69-78. DOI: https://doi.org/10.5713/ajas.13.0833):

Breeding weight is the target. The animals in your study were fed to attain 70% mature BW at the beginning of the breeding season, which follows Freetly HC, Kuehn LA, Cundiff LV (2011, Growth curves of crossbred cows sired by Hereford, Angus, Belgian Blue, Brahman, Boran, and Tuli bulls, and the fraction of mature body weight and height at puberty. J Anim Sci 89:2373–2379) that attainment of an absolute BW, or age, was less critical than developing heifers across breeds to a proportion of mature BW. Further, Kasimanickam RK, Kasimanickam VR, McCann ML. Difference in Body Weight at Breeding Affects Reproductive Performance in Replacement Beef Heifers and Carries Consequences to Next Generation Heifers. Animals (Basel). 2021 Sep 26;11(10):2800. doi: 10.3390/ani11102800. PMID: 34679822; PMCID: PMC8533008.

  1. In conclusion, 55% of MBW at breeding negatively affected the reproductive performance of heifers and its offspring heifers. The recommendation is to feed heifers a balanced diet to reach 65% of MBW at breeding with consideration of production traits.
    1. Thank you for your comment. The heifers in thist study were developed to 70% of the estimated mature target body weight as stated in line 124. This was due to the fact these heifers were from multiple herds where cow size was not documented in the development yard.  While you make an excellent point in the ongoing debate of whether or not to develop heifers to 55% or 65%, this article would in fact support the 65% claim as you were making the point for (line 271-272).
  2. The cattle in your study are of various breeding.  So, would it not be of more importance for the modeling to be based on the percentage of mature BW rather than kg.
    1. Thank you for your comment. The heifers were from different farms of various breeds and herd sizes, the percentage of mature BW would be simply speculation or a guess at this point, based on the no breeds recorded in this data set. Therefore, we could not calculate nor, include it and used breeding weight instead.

L 21 “ to discover traits and management practices to eliminate subfertile heifers prior to breeding” implies indirect selection for the counter: fertile heifers. Your reply-

The objective of the study was to conduct a retrospective observational study to identify risk factors. Retrospective studies are useful to generate hypothesis by identify factors that can be investigated later. The objective of the study was not to build a prediction model. In addition, when building prediction models, you always need the outcome to be able to train the model, so you need the result of the outcome either way.

None of the risk factors that you examined is novel (e.g. Moorey, S.E., Biase, F.H. Beef heifer fertility: importance of management practices and technological advancements. J Animal Sci Biotechnol 11, 97 (2020). https://doi.org/10.1186/s40104-020-00503-9). Further,  a comparable study, Dickinson SE, Elmore MF, Kriese-Anderson L, et al. (Evaluation of age, weaning weight, body condition score, and reproductive tract score in pre-selected beef heifers relative to reproductive potential. Journal of Animal Science and Biotechnology. 2019 ;10:18. DOI: 10.1186/s40104-019-0329-6. PMID: 30891236; PMCID: PMC6390375), which you cited, had reported

  1. Here, we tested the hypothesis that in a group of pre-selected heifers, records of weaning weight, age at weaning, age at artificial insemination, and age of dam differ among heifers of varied reproductive outcomes during the first breeding season. None of the parameters tested presented predictive ability to discriminate the heifers based on the response variable ('pregnant to artificial insemination', 'pregnant to natural service', 'not pregnant'). Heifers younger than 368 d at the start of the breeding season did not become pregnant to artificial insemination. Those young heifers had 12.5% chance to become pregnant in their first breeding season, compared to 87.5% if the heifers were older than 368 days.
    1. Thank you for your comment. From lines 77-82 of the manuscript: “This study utilized production records from 2,416 heifers housed at a private heifer development yard in Iowa. This facility develops heifers in a confinement environment with both covered and open pens with concrete flooring. There were 78 different cattle sources represented in this data set with heifers born over a four-year period (2013 to 2017). Prior to delivery to the yard, each individual producer selected the heifers for development based on their operation’s needs and selection criteria.”  The study cited above (Moorey et al) was done in the southeastern United States using records from a relatively small number of heifers (n= 259) and produced from a consistent university breeding program (all Angus x Simmental). As expected, these heifers were developed primarily on grazed forage with minimal supplementation on an as needed basis – both pre- and post-breeding. Our study is vastly different than the conditions described in Moorey et al (number of animals, location, developed in a feedyard setting, nutritional management with a TMR, bred for spring calving, etc.). Due to economic factors in the midwestern United States, custom heifer development in a dry lot setting is becoming more common and financially feasible for both producers and the development yard. The type of research and analysis done in this paper is valuable to producers and veterinarians as it helps to prioritize how heifers should be developed – both prior to and while at the development yard. Whether the outcomes agree with previous publications is not central to the value and relevance of this research. The value resides in the fact that the results are applicable to heifers developed in the largest cattle producing area of the world using a novel production system. In fact, one could argue that this research helps strengthen the body of literature that deals with different heifer development systems in different parts of the United States. Therefore, the authors choose to respectfully disagree with the reviewer.